# Zika Virus

**DOI:** 10.3390/pathogens9110898

**Published:** 2020-10-28

**Authors:** Sophie Masmejan, Didier Musso, Manon Vouga, Leo Pomar, Pradip Dashraath, Milos Stojanov, Alice Panchaud, David Baud

**Affiliations:** 1Maternofetal and Obstetrics Research Unit, Department “Woman-Mother-Child”, University Hospital, 1011 Lausanne, Switzerland; sophie.masmejan@chuv.ch (S.M.); manon.vouga@chuv.ch (M.V.); leo.pomar@chuv.ch (L.P.); Milos.Stojanov@chuv.ch (M.S.); 2Laboratoire Eurofins Labazur Guyane, 97300 Cayenne, French Guiana; dmusso12345@gmail.com; 3Aix Marseille University, IRD, AP-HM, SSA, VITROME, IHU-Méditerranée Infection, 13007 Marseille, France; 4Division of Maternal-Fetal Medicine, Department of Obstetrics and Gynecology, National University Hospital, Singapore 119074, Singapore; pradip_dashraath_vijayakumar@nuhs.edu.sg; 5Service of Pharmacy, Lausanne University Hospital and University of Lausanne, 1011 Lausanne, Switzerland; alice.panchaud@chuv.ch; 6Institute of Primary Health Care (BIHAM), University of Bern, 3012 Bern, Switzerland

**Keywords:** Zika, Zika virus, emerging infectious diseases, congenital Zika syndrome, materno-fetal infections, Guillain-Barré syndrome

## Abstract

Zika virus (ZIKV), a neurotropic single-stranded RNA flavivirus, remains an important cause of congenital infection, fetal microcephaly, and Guillain-Barré syndrome in populations where ZIKV has adapted to a nexus involving the *Aedes* mosquitoes and humans. To date, outbreaks of ZIKV have occurred in Africa, Southeast Asia, the Pacific islands, the Americas, and the Caribbean. Emerging evidence, however, suggests that the virus also has the potential to cause infections in Europe, where autochtonous transmission of the virus has been identified. This review focuses on evolving ZIKV epidemiology, modes of transmission and host-virus interactions. The clinical manifestations, diagnostic issues relating to cross-reactivity to the dengue flavivirus and concerns surrounding ZIKV infection in pregnancy are discussed. In the last section, current challenges in treatment and prevention are outlined.

## 1. Introduction

Zika Virus (ZIKV) is an RNA *Flavivirus* which was first isolated in 1947 in Uganda, Africa [1]. Its impact on public health was limited until the first viral outbreak in the Yap Islands, Pacific, in 2007 [2], followed by its emergence in French Polynesia in 2015 and subsequently in other Pacific Islands [3] and its massive spread into the Americas from 2015. The severe consequences of ZIKV infection in pregnancy, such as microcephaly and severe neurological impairment of the newborn, among other issues, were then rapidly identified [4]. On February 2016 Zika (the disease caused by ZIKV) was declared to be a public health emergency of worldwide importance by the World Health Organization (WHO) [5]. Since 2017, the transmission of ZIKV declined; ZIKV, however, is still circulating and could potentially cause massive outbreaks in some regions [6]. The emergence of ZIKV in South America has been extensively reported in the literature and readers can report to previous reviews [6,7], thus, in this review, we discuss the 2019–2020 recent findings in epidemiology, transmission, immunogenicity and host factors, clinical features and potential treatments of Zika.

## 2. Epidemiology

ZIKV is a single-stranded RNA virus belonging to the *Flavivirus* genus in the *Flaviridae* family. Its RNA encodes for a polyprotein which is then cleaved into capsid, envelope, pre-membrane, and non-structural proteins. ZIKV produces untranslated RNA that could be involved in regulating host antiviral response and inducing host cell death [8].

After its discovery in Uganda in 1947 (ZIKV Asian African lineage) [1], ZIKV spread to Asia in the 1960s (ZIKV Asian lineage) but only sporadic cases were described prior to 2007, when the first Zika outbreak occurred in Yap Islands, Micronesia, Pacific, infecting 75% of the population [2]. In 2013–2014, a second Zika outbreak occurred in French Polynesia, Pacific, where more than 50% of the population was infected [9]. ZIKV then spread to other Pacific Islands and to South America where it was first detected in north eastern Brazil as of 2015 [3,10]. Severe complications from Zika were reported during these emergences; Guillain-Barré Syndrome when ZIKV emerged in French Polynesia, and severe complications in newborns from infected mothers when ZIKV emerged in Brazil (also retrospectively reported in French Polynesia [9]). Complications in newborns where subsequently described as “congenital ZIKV syndrome” [11], with more than 3,700 cases reported in Brazil as of January 2018 [6]. ZIKV is now considered the newest member of the TORCH pathogens [12]. Although no evidence for universal screening of ZIKV with other TORCH pathogens has been found yet, the similarities between TORCH infections and ZIKV infections are striking, especially because of their neurotropism, and their association with microcephaly. Cytopathic effect induced by ZIKV in brain is similar to Rubella’s. In addition, severity of fetal infection, transmission rate and clinical manifestations depend on the gestational age of the infection in TORCH infections as well as in ZIKV infections [13]. ZIKV is thus a considerable threat to the fetus when a prenatal exposure occurs [14]. After this massive pandemic, since 2017, the incidence of Zika has considerably decreased. The most common hypothesis to explain the decline of the circulation of ZIKV is a high herd immunity of the population. However, herd immunity can only explain the decrease of ZIKV circulation in regions where the majority of the population have been infected, such as French Polynesia and Northeast of Brazil. The decrease of the epidemy in the rest of the world is still a matter of debate [2,15,16]. 

As of July 2019, 87 countries and territories in Africa, Asia, the Americas and Pacific have reported the presence of Zika [17]. Additional cases of microcephaly were detected in several African countries at that time, for instance in 2017 in Angola, 42 cases of infants with microcephaly were reported. In 2015–2016 in Cabo Verde, 7580 infants were suspected to have congenital ZIKV infection, of which 18 had microcephaly. In Guinea-Bissau, in 2016, 15 infants with microcephaly were reported of which 13 had a confirmed ZIKV infection. The association between microcephaly and ZIKV infection in these cases remains although unclear mainly because of a lack of surveillance of the infection in these countries. Sixty-one countries and territories, however, have not yet documented autochthonous Zika vector borne transmission, despite the presence of endemic *Aedes aegypti*, the main ZIKV mosquito vector [17]. 

Autochtonous Zika vector borne transmission has never been reported in Europe prior to 2019 despite more than 2,300 imported cases reported between 2015 and 2017 [18]. Surprisingly, the first locally acquired infections were reported in autumn 2019, in southern France, a region where *Ae. aegypti* is absent and where the potential mosquito vector is *Aedes albopictus* [19]. These new data suggest that ZIKV has the potential to emerge in all countries with potential mosquito vectors (*Aedes* species), and of course to re-emerge in countries already impacted in the past [20].

As the circulation of the virus is unpredictable, surveillance and research on the emergence of arboviral diseases should be increased, due to the worldwide threat they represent in an increasingly globalized world [21].

ZIKV emerged on a large scale in the Americas where it has been extensively reported [6,7]. Interestingly, several outbreaks have been reported in Asia but of limited importance and ZIKV also failed to emerge on a large scale in Africa. 

The reason ZIKV failed to emerge on a large scale in Africa, which is where it originated, is unclear. Several historical studies have reported the isolation of ZIKV in non-human primates and sylvatic mosquitoes in Africa and the detection of antibodies against ZIKV in humans and other mammals [22]. The first reported ZIKV outbreak in Africa occurred in Cabo Verde in 2015 but was caused by a strain belonging to the Asian lineage, imported from Brazil [23]. Two other outbreaks occurred in Angola and in Gabon with strains belonging to the African lineage. Cases of microcephaly were reported in these three outbreaks; however, the link between ZIKV and fetal birth defects has not yet been demonstrated in Africa. Recent serological studies suggest that ZIKV may continue to circulate in an undetected manner in Africa. From 574 samples collected in 2013 in Kenya, five had confirmed positive IgM for ZIKV [24]. A seroprevalence study conducted between 2007 and 2012 in Senegal, Mali and Gambia, estimated a seroprevalence around 20–22% [25]. Altogether, these data confirm that ZIKV has been circulating for a long time in Africa, but for unknown reasons, has never been responsible for any large outbreak in this continent. In vitro and animal studies stress this paradox. In mouse embryos, ZIKV African strains are capable of causing as many birth defects and fetal losses as the Asian strain [26]. Similar results were observed in pigs, with high viral loads present in the placentas, membranes and embryos, leading to microcephalic fetuses [27]. In general, animal and in vitro studies suggest a high pathogenicity of the African lineage compared to the Asian lineage. The link between the ZIKV African lineage and microcephaly or other birth defects, however, has not been demonstrated in human populations. 

The situation is different in Asia where ZIKV has been known to circulate since the 1960s but the first outbreak was reported only in 2016 in Singapore [28]. Several other outbreaks have been subsequently reported [29]. ZIKV also emerged in India in 2018 but with less than 300 reported cases [30]. 

It should be noted that ZIKV outbreaks in Asia were caused by strains belonging to the “old Asian lineage” and not by the strains belonging to the “American or emerging lineage”, and that cases of microcephaly have been reported as a consequence of infections by both of these Asian lineages [31].

The next epidemic could potentially occur in Europe, as exemplified by the autochthonous cases that were identified in France in 2019 [32]. The primary potential mosquito vector in Europe, however, is *Ae. albopictus* which is responsible for chikungunya virus infection but is unlikely to be able to sustain large ZIKV outbreaks. Indeed, *Ae. aegypti* exhibit a much higher transmission potential for ZIKV than *Ae. albopictus* [33,34,35].

## 3. Transmission 

Transmission of ZIKV can be mosquito-borne, sexual, transfusion-based, or vertical (materno-fetal).

### 3.1. Mosquito-Borne Transmission

The primary mode of transmission of ZIKV is mosquito-borne transmission with *Ae. aegypti* being the major vector for ZIKV [36]. According to mathematical models, *Ae. aegypti* presence could continue to expand due to population growth, movement and climate change [19]. *Ae. aegypti* is present in all tropical regions of the world, placing those at highest risk of new outbreaks. The widely distributed *Ae. albopictus* can also transmit ZIKV but seems to play a minor role in the recent outbreaks, it was reported as the potential vector only in Gabon in 2007 [37] and recently in sporadic cases in France [20]. *Culex* species, among others, have been reported to be carriers of ZIKV but their ability to transmit the virus to humans seems nonexistent [38,39]. Other *Aedes* species with limited geographic distributions can also transmit Zika, for instance *Aedes hensilli* on Yap Island or *Aedes polynesiensis* in French Polynesia [40,41].

### 3.2. Vertical Transmission

Vertical transmission from mother to child can occur and cause severe congenital malformations as noted previously. Maternofetal transmission of ZIKV is estimated to occur in 20% to 30% of infected pregnant women [6,42,43]. Timing of infection and its impact on fetal transmission remain difficult to establish [43]. Infections in the first trimester of pregnancy, however, are at higher risk of Congenital Zika Syndrome (CZS) [44]. 

Presence of ZIKV RNA in breastmilk has been described in case reports [45]. However, ZIKV transmission via breast feeding has never been demonstrated and, according to WHO, breast feeding is not contraindicated for ZIKV-infected mothers [6].

### 3.3. Sexual Transmission of ZIKV and the Effect of ZIKV Infection on Male Fertility 

ZIKV sexual transmission has been suspected since 2008, when a traveler returning from an endemic area infected his partner who was in a region where vectors were absent [46]. Although potential sexual transmission of arboviruses other than ZIKV has been suspected, it has never been demonstrated [47]. ZIKV thus appears to be the only arbovirus that can be sexually transmitted [6]. The frequency of sexual transmission is difficult to estimate as all of the population is exposed to the vector in endemic regions. Male-to-female, female-to-male and male-to-male transmissions have been described [48]. In men, ZIKV RNA shedding was demonstrated in two thirds of semen samples tested within 30 days of illness onset, which decreased substantially within a month. The maximal persistence of ZIKV RNA was 281 days in one patient [49]. ZIKV RNA can persist for several months in semen, but infective particles seem to be limited to the first weeks of illness [49,50], making the presence of ZIKV RNA an unreliable indicator of the presence of infectious ZIKV. The presence of ZIKV RNA in vaginal secretions is rare (around 2%) and its persistence is thus difficult to assess [50]. Recommendations of the Centers for Disease Control and Prevention regarding preconception counseling are that men should use condoms for 3 months after returning from endemic areas (or after the last possible ZIKV exposure) and that women wait for two months before trying to conceive [51]. These recommendations seem reasonable as the longest interval between infection and sexual male to female transmission ever reported is 44 days with a peak infectivity at 2 weeks [52]. The risk of intrauterine transmission among ZIKV-infected women trying to conceive near the end of the recommended 8 week period is thought to be small as 95% have no detectable ZIKV RNA in serum after 6 weeks from disease onset [50]. 

The effect of Zika on male fertility is a matter of concern. The testis is an immunoprivileged organ, protected by the blood-testis barrier in order to protect spermatogenesis. Despite this mechanism, pathogens such as ZIKV have the ability to persist in the male genital tract. The consequences of this on male fertility need to be determined. Acute Zika may alter the quality of sperm with evidence of a decreased sperm count between day 7 and day 60 after infection. Total motile sperm count decreased by 50% at day 60. Inhibin values increased until sperm count recovered (after 120 days). Persistence of RNA was detected in 3 out of 15 patients at day 120. The presence of the virus in the semen was associated with the reduced sperm count [53]. Sertoli cells, which are responsible for protection of the spermatogenesis by forming the blood-testis barrier, support ZIKV replication, and their transcriptional profile can be significantly modified by ZIKV infection. ZIKV infection can promote cell death in spermatogonia and thus damage the reproductive male system [54,55]. These findings, together with a mouse model study suggest that ZIKV could lead to male infertility; however, more studies are needed to confirm this statement [56]. 

### 3.4. Transmission through Blood Transfusion

Given the presence of ZIKV in blood donors, and the report of four possible cases of transfusion-associated transmission of Zika, Zika should be classified as a potential transfusion-transmitted disease [57]. As it was the case for sexual transmission, the burden of this mode of transmission in endemic areas is difficult to assess. The prevalence of ZIKV RNA in blood donors was estimated to be around 1% [58]. This emerging disease is a challenge for blood banks, and strategies should be implemented in order to prevent transmission by blood transfusion. Currently, the USA has adopted testing of all blood donations for ZIKV RNA with exclusion of positive donations, even in regions without *Aedes* mosquitoes or ZIKV cases. Another strategy could be to defer blood donation after traveling to endemic areas; however, this would require an increase in the pool of available donors [59]. Pathogen inactivation is also an effective solution but is only commercially available for plasma and platelet blood products [60]. Strategies in endemic areas are more challenging as testing is an expensive strategy for low-income countries. Selective inventory for pregnant women could be an option [61]. The most cost-effective strategy should be individualized for each country.

### 3.5. ZIKV in Solid Organ Transplanted Patients

Transmission by solid organ transplantation is also a suspected mode of transmission. US guidelines recommends that the donation should be deferred when there is a suspected case of ZIKV-positive living donor. When this strategy is not possible, concerns should focus on benefits of the transplant versus the risk of severe infection of the recipient [62]. However, to our knowledge, ZIKV transmission by transplantation has never been firmly confirmed.

### 3.6. Contact with Infected Body Fluids

ZIKV can be detected in urine [63] and saliva [64]. Although, as discussed further, the method of choice for diagnosis is the RT-PCR in blood, diagnosis in saliva is also possible and can be useful when blood sampling is challenging [64]. Interhuman transmission however has only been suspected once. This way of transmission has thus not been proven yet [65]. 

## 4. Clinical Features

### 4.1. Zika in Fetuses and Infants

Infection of the fetus can result in severe malformations. CZS is a combination of severe neurological anomalies including structural brain anomalies, fetal hydrops, arthrogryposis, ocular anomalies, and clinical signs such as microcephaly, hypertonia, swallowing disorders, following in utero exposure to ZIKV [66]. Complications, however, are not limited to CZS. Among women infected during pregnancy, CZS occurs in 5 to 14% and microcephaly in 4 to 6% [6]. The developmental outcome of children with CZS is extremely poor with nearly 100% of children presenting with severe developmental injuries, as described at a mean age of 30 months [67]. Complications are reported even in children without any clinical or radiological abnormalities at birth. The motor scores of children with prenatal exposure to ZIKV without microcephaly was significantly lower than those of controls, suggesting that even without evident congenital ZIKV symptoms, a prenatal exposure to ZIKV can have serious developmental consequences [68]. Another recent study suggests that among children with normal neurological evaluation at 6 to 12 months, the rate of developmental injuries was higher in children with prenatal exposure to ZIKV in the third trimester than in the first trimester of pregnancy [69]. This is contradictory to the known short-term outcomes stating that first trimester infections have a worse developmental prognosis compared to third trimester infections [70]. Long-term studies are still needed to evaluate the rates of complications in the medium and long term following birth of ZIKV-infected children [71].

### 4.2. Zika in Children

WHO and Pan American health Organization (PAHO) have issued clinical diagnostic criteria for Zika. The WHO case definition of Zika includes rash or fever and at least one of the following: arthralgia, arthritis, conjunctivitis [72]. PAHO criteria include rash and at least 2 of: fever, conjunctivitis, arthralgias, myalgia and periarticular edema [73]. Based on evaluation of 556 ZIKV-positive children aged 2 to 14 years old, children tend to show mild clinical findings compared to adults. Indeed, only 32% of them met the WHO clinical diagnostic criteria, and 20% of them met the PAHO clinical diagnostic criteria. Children are also less frequently affected by arthralgia, regardless of their ability to communicate the presence of this symptom [74,75]. Indeed, most children resented with only a rash or a rash with leucopenia, which is a clinical presentation that does not meet the WHO or PAHO criteria. The sensitivity of the criteria improves with age [74]. With a few exceptions, severe complications have not been reported in children.

### 4.3. Zika in Adults

Most ZIKV infections are asymptomatic (75–80%). After an incubation period of 3 to 14 days, clinical manifestations include rash, low-grade fever, arthralgia, myalgia and conjunctivitis [22]. Neurological complications resulting from the neurotropism of the virus can occur: meningoencephalitis, myelitis or Guillain-Barré syndrome [76,77,78]. Guillain-Barré syndrome has a prevalence of 2 to 3 per 10,000 ZIKV infection [79], and is suspected to cause a higher rate of morbidity and cranial neuropathy compared to other etiologies [6]. 

Persistent shedding of ZIKV RNA in patients infected by ZIKV has been well described [80,81,82]. However, renal pathology, if present, remains subclinical at least in immunocompetent patients.

## 5. Diagnosis

During the symptomatic phase, ZIKV Reverse Transcription PCR (RT-PCR) in blood is the exam of choice [83]. ZIKV can be detected in serum and whole blood [84]. ZIKV RNA is also detectable for more than 10 days in urine [63,85]. Of note, ZIKV RNA can sometimes be detected for a longer period during pregnancy (a 3-fold longer estimated median detection time of ZIKV RNA in serum was assessed among a cohort of 18–39 year old pregnant women in Puerto Rico), although detection can be intermittent [86,87,88]. 

Specific antibodies against ZIKV can be detected from the second week post symptom onset (IgM) using “in house” or commercially available ELISA. This detection, however, is limited by cross reactions with other flaviviruses, principally DENV, resulting in false positives [89]. All positive ZIKV ELISA results should be confirmed using the reference serologic assay, plaque reduction neutralization (PRNT) [90]. Unfortunately the availability of PRNT assays is restricted to highly specialized laboratories and is also not completely exempt from cross reactions [91].

Even when performed correctly, however, a negative ZIKV RT-PCR within the week post infection or negative serologic evaluation from the second week post infection does not rule out the diagnosis. In addition, it is impossible to determine the timing of infection in asymptomatic pregnant women using laboratory testing.

## 6. Host-Virus Interactions

ZIKV has a tropism for a broad range of organs such as the brain, placenta, testis, and retinal cells. ZIKV can be detected in body fluids such as blood, amniotic fluid, semen, vaginal secretions, tears, urine or saliva [92]. The tropism of ZIKV for neuroprogenitor cells, which results in induction of cell cycle arrest, apoptosis, and arrest of differentiation, explains the severe microcephaly described in CZS. ZIKV is able to overcome the defenses of the placenta against pathogens by entering placental macrophages [93,94,95], and thus reach the fetus. In addition to CZS, the susceptibility of trophectoderm cells for ZIKV could prompt first trimester loss or alter neural development in the very early stages of the pregnancy [96]. The tropism of ZIKV for certain tissues can be explained by a specific receptor present on target cells, encoded by the gene *AXL* [97]. Ocular complications have also been identified in fetuses and infants exposed in utero, with a broad spectrum of anomalies: from macular scars to microphthalmy. 

ZIKV is able to modify cell death pathways to induce apoptosis [8]. Multiple cellular death pathways induced by ZIKV have been identified including apoptosis, necrosis and paraptosis [98,99]. Further understanding of these pathways could help specific drug development. 

### 6.1. Host Immune Response

Primary immune response against ZIKV is orchestrated by interferons I, II, and III, which activate Interferon-stimulated-genes that target the virus life cycle and stop the infection [8]. Interferon-inducible transmembrane proteins (IFITMs) are known to inhibit the replication of various flaviviruses such as West-Nile (WNV) and dengue (DENV) viruses. This confers a hope for the development of new anti-ZIKV medications as IFITMs can also inhibit ZIKV in human cells [100].

ZIKV, on the other hand, restricts the host immune response. As an escape mechanism, it uses its non-structural proteins to inhibit IFN I. Non-structural proteins are also able to promote autophagy in brain host cells resulting in enhanced viral replication [8].

### 6.2. Complex Interactions between Anti DENV and ZIKV Immunity

Adaptive host immune response consists of humoral and cellular responses. The humoral response generates neutralizing antibodies that recognize different domains of the E protein. A secondary DENV infection can be exacerbated by a phenomenon called antibody-dependent enhancement (ADE) [101,102,103]. ADE is a challenge in endemic areas because patients can be successively infected by different DENV serotypes [104]. The consequences of a previous DENV infection on ZIKV infection and vice versa; however, are unknown. Conflicting results from both in vitro and in vivo studies exist and to date it is impossible to state if a previous infection by ZIKV or DENV protects from or enhances a secondary infection by the heterologous *Flavivirus* [105,106]. 

For example, it was found that the concentration of DENV-NS1 IgG antibodies was inversely correlated with the probability of ZIKV infection. DENV immunity could thus have a protective effect against subsequent ZIKV infection. In Salvador (South America) in 2015, 73% of the population was infected with ZIKV, and the presence of preexisting DENV immunity was suspected to have potentially lowered the risk of ZIKV infection. This statement could also explain the low incidence of ZIKV infection in Asia which is a DENV endemic region [16,107]. In French Polynesia, however, more than 80% of the population had antibodies against DENV before the emergence of ZIKV and which then infected more than 50% of the population [15,108]. 

With regards to vaccine development, a previous DENV or ZIKV infection can be a challenge as it can predispose an individual to a more severe dengue or Zika infection [104]. ZIKV and DENV are closely related viruses with a high sequence identity for the structural proteins capsid and envelope (44–56%) as well as for the more conserved non-structural proteins NS3/NS5 (68%), which represent the main targets of the CD4+ and CD8+ T-cell response to DENV, respectively [109]. Similar to ADE driven by antibodies, DENV-elicited T cells are cross-reactive with ZIKV [110,111].

## 7. Prevention and Treatment

Over the past few years, the scientific community has made considerable efforts to develop a vaccine and bring it to licensure for public use. Studies on rhesus macaques suggest that the adaptative immune response induced by ZIKV is protective against re-challenge by the homologous virus [112], raising the hope for the potential to development a ZIKV vaccine. Different vaccine candidates have been tested in vitro, in vivo, and some progressed to clinical trials. The different candidates include DNA vaccines expressing pre-membrane and envelope proteins, inactivated virus vaccine, recombinant protein subunit vaccine, lipid nano-particle encapsulated mRNA vaccines, virus-like particles and live-attenuated vaccines [14]. This is encouraging; however, no candidate has been licensed yet and several challenges still need to be addressed before an effective vaccine can be brought to the market. For example, the reduction in Zika cases makes vaccine efficacy trials in the absence of an outbreak difficult to perform [104,113].

To date, there is no specific antiviral drug targeting ZIKV. Several new therapeutic candidates and existing drugs have been screened and proposed to target ZIKV via different mechanisms. Direct-acting antiviral drugs, including nucleoside analogs and polymerase inhibitors, target the RNA-dependent RNA polymerase (RdRp). Drugs that target host cell processes are directed against any of the different steps of the viral life cycle, such as purine or pyrimidine synthesis inhibitors or ZIKV-entry inhibitors. Several drugs have shown anti-ZIKV activity in vitro and in animal models; however, to date, none have entered into clinical trials and shown any evidence of anti-ZIKV activity in humans [114].

## 8. Conclusions

The dramatic consequences of ZIKV on public health since 2015, have highlighted the threat that ZIKV represents. Although the pandemic has waned since that time, the virus is still circulating, and areas with competent vectors are at risk of ZIKV re-emergence. An estimated 3.6 billion people live in at-risk areas [115]. Several challenges still need to be addressed such as the development of specific drugs and vaccines, the need for improved epidemiological surveillance in at-risk areas, and the determination of long-term consequences of ZIKV infection, in particular in cases of prenatal exposure.

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
