# Peer review of "Zika Virus"

_pathogens, 2020, doi:10.3390/pathogens9110898_

Round 1

Reviewer 1 Report

Dear Author,

The manuscript describes how

Overall: In my opinion the manuscript is highly abbreviated in scope having short sections lacking in details and with no references. The biology and pathogenesis of the virus is also abbreviated. This type of review usually requires a more robust overview of virus biology, pathogenesis, immunity, and treatment strategies. After my review, this manuscript should undergo major revisions based on my comments below before being considered for publication.

Major Comments

  1. The introduction section is highly abbreviated in depth with only a single reference. There is no detailed information in the introduction that is highly warranted in this type of review. The introduction needs to be expanded in scope.

  1. There are grammatical errors in manuscript that should be addressed.

  1. The manuscript is poorly referenced.

  1. Example: “In 2013-2014, a second Zika outbreak occurred in French Polynesia, Pacific, where more than 50% of the population was infected” there is no reference.

  1. The statement “ZIKV is now considered the newest member of the TORCH pathogens and a considerable threat to the fetus when a prenatal exposure occurs.” Is there any indication for universal TORCH screening for ZIKV. This should be address in the manuscript.

  1. The statement: “As of July 2019, 87 countries and territories in Africa, Asia, the Americas and Pacific have reported the presence of Zika.” Needs a reference. The incidence of microcephaly as of 2019 in these regions should be stated here along with virus lineage.  

  1. The statement: “Asia but of limited importance and ZIKV also failed to emerge on a large scale in Africa. We will focus on these two regions.” The rationale for focusing on these two regions and excluding Brazil/South American is unclear.

  1. There is no mention of ZIKV shedding in saliva and urine as a viral reservoir and as a means of transmission.

  1. The statement: “The presence of ZIKV RNA in vaginal secretions is rare (around 2%) and its persistence is thus difficult to assess.” There is no reference for this important statement.

  1. ZIKV transmission in transplant populations needs to be included.

  1. The statement: “Most ZIKV infections are asymptomatic (75-80%). After an incubation period of 3 to 14 days, clinical manifestations include rash, low-grade fever, arthralgia, myalgia and conjunctivitis. Neurological complications resulting from the neurotropism of the virus can occur: meningoencephalitis, myelitis or Guillain-Barré syndrome. There are no references for these statements.

  1. There is no mention of ZIKA infection of the kidney or oral cavity.

  1. The statement: “ZIKV is able to overcome the defenses of the placenta against pathogens by entering placental macrophages, and thus reach the fetus.” There is no reference for this statement.

  1. Most people infected with ZIKV including children will have mild disease due to a robust antibody response. That needs to be made clear.

  1. The statement “DENV infection can be exacerbated by a phenomenon called antibody-dependent 255 enhancement (ADE).” There is no reference for this statement.

  1. The statements: “For example, it was found that the concentration of DENV-NS1 IgG antibodies was inversely correlated with the probability of ZIKV infection. DENV immunity could thus have a protective effect against subsequent ZIKV infection. In Salvador (South America) in 2015, 73% of the population was infected with ZIKV, and the presence of preexisting DENV immunity was suspected to have potentially lowered the risk of ZIKV infection.” There are no references for these statements.

  1. There are several ZIKV vaccines that have been developed but none FDA approved. This is not discussed.

  1. The author stated the manuscript would be focus on 2019 however it was limited to 2 regions but there was very little discussion on these 2 regions.
  2. There are no figures in the manuscript.

Reviewer 2 Report

This review briefly discusses several aspects of ZIKV including epidemiology, transmission, clinical features of disease and lightly touches on host-virus interactions. It provides a nice snapshot of these topics as they pertain to ZIKV. However, there are several sections that are a bit disjointed and disrupt the flow for the reader. In addition, the authors do not discuss cellular immunity within context of ZIKV. These and other comments are provided below.

  1. lines 57-75 are disjointed, please consider re-writing
  2. line 55 - any reason as to why the “incidence of ZIKV has considerably decrease”
  3. line 75 – can the authors justify why they are focusing on 2 geographical regions?
  4. what is the point of mentioning the statements in line 99-102? Can the authors please elaborate?
  5. when authors discuss vectors (line 105, 106) they mention one that is “unlikely to be able to sustain large ZIKV outbreaks”. Any reference for this? Why is this vector not suitable for causing large outbreaks of ZIKV?
  6. line 112, authors mentioned “models”; can the authors specify what models they mean here? Also, include a reference.
  7. Please define abbreviations throughout the text. Several locations were noted where abbreviations were not defined at first mention. For example, CZS (line 123) defined in line 177; A. aegyptii not defined prior to its usage.
  8. line 117 the authors mention “other Aedes species” - what other species are they referring to and how many other species exist?
  9. line 125, it is “the World Health Organization” not “World Health Organization”.
  10. line 127, change to sexual transmission of ZIKV…
  11. line 197. Please modify the sentence so that you do not use “:” twice in 1 sentence. This is not grammatically correct.
  12. line 214. How many days post infection do you mean by first?
  13. line 126: “longer time in urine”, how many days longer, same comment for pregnancy.
  14. line 233 “consequence of brain disruption sequence” what does this mean?
  15. section 6.2, line 270-271 – authors mention cellular response which mainly involves CD8+ T and CD4+ T cells. This is a very general statement and lacks specific details that pertain to ZIKV infection. Are there no additional insights to this process in terms of ZIKV infection? Please expand on this section.
  16. section 7. The first two paragraphs that discuss vaccines are very redundant and somewhat contradictory. Authors in first paragraph suggest there is a vaccine candidate but second paragraph states there is not a vaccine candidate. Please re-write these for clarity and flow.
